# Assessing Digital Soil Inventories for Predicting Streamflow in the Headwaters of the Blue Nile

**Anwar A. Adem** [1,2], **Yihun T. Dile** [3], **Abeyou W. Worqlul** [4], **Essayas K. Ayana** [3],
**Seifu A. Tilahun** [1] **and Tammo S. Steenhuis** [1,5,*]

1   Faculty of Civil and Water Resources Engineering, Bahir Dar University, Bahir Dar 6000, Ethiopia
2   Department of Natural Resource Management, Bahir Dar University, Bahir Dar 6000, Ethiopia
3   Texas A&M Univ, College Station TX 77843, USA
4   Texas A&M AgriLife Res, Temple, TX 76502, USA
5   Biological and Environmental Engineering, Cornell University, Ithaca, NY 14853, USA
*   Correspondence: tss1@cornell.edu; Tel.: +1-607-255-2489

**Abstract:** Comprehensive spatially referenced soil data are a crucial input in predicting biophysical and hydrological landscape processes. In most developing countries, these detailed soil data are not yet available. The objective of this study was, therefore, to evaluate the detail needed in soil resource inventories to predict the hydrologic response of watersheds. Using three distinctively different digital soil inventories, the widely used and tested soil and water assessment tool (SWAT) was selected to predict the discharge in two watersheds in the headwaters of the Blue Nile: the 1316 km$^2$ Rib watershed and the nested 3.59 km$^2$ Gomit watershed. The soil digital soil inventories employed were in increasing specificity: the global Food and Agricultural Organization (FAO), the Africa Soil Information Service (AfSIS) and the Amhara Design and Supervision Works Enterprise (ADSWE). Hydrologic simulations before model calibration were poor for all three soil inventories used as input. After model calibration, the streamflow predictions improved with monthly Nash–Sutcliffe efficiencies greater than 0.68. Predictions were statistically similar for the three soil databases justifying the use of the global FAO soil map in data-scarce regions for watershed discharge predictions.

**Keywords:** soil database; soil survey; soil resource inventory; SWAT; watershed; erosion; sediment; Ethiopia; Ethiopian highlands; Africa

---

## 1. Introduction

Soil data are crucial for landscape and water-resource planning [1]. With the advancement of remote-sensing technologies, geostatistics and geographic information system (GIS) data integration, soil datasets have become available in digital form with ever increasing precision and utility. The outcome of hydrological models is strongly influenced by spatial variability of ecological and physical processes in the landscape, which are linked with soil genesis [1–4]. Adequate representation of soil genesis underlying the soil classification digital soil inventories has become especially important in hydrological modeling [1]. Unfortunately, in developing countries, many digital soil inventories are not easily and freely available [5]. So, before investing in a digital soil inventory, there is a need to assess the accuracy required for the soil inventory for improved simulation results.

Lumped and distributed hydrological models and combinations of these two (hybrid models) are important in predicting stream flow [6,7]. Lumped models use mean effective map values of slope, soils and land use to simulate discharge [8–10], that assess the catchment response simply at the outlet without considering the spatial distribution of the input parameters [10]. Distributed models that divide up the landscape in modeling units require spatial map data as input for these modeling units [10–13].

The distributed models require more data for parameterization than the lumped models [11]. Different representation of soils in the modeling units may affect watershed heterogeneity during watershed discretization of distributed models [14]. Such differences in representation become a source of input data uncertainty in hydrologic modeling [15–19] which may lead to output uncertainty [20].

Most research on evaluating the effect of soil inventories and variability on streamflow predictions [21–30] has been carried out in the United Sates comparing the State Soil Geographic (STATSGO)– 1:15,000 to 1:31,000 and Soil Survey Geographic (SSURGO)–1:250,000 soil databases. For example, Prasanna and Mulla [25] showed that STATSGO soil data were slightly better in predicting stream flow than SSURGO soil data even when STATSGO's resolution was coarser than that of SSURGO. Similar findings were obtained by Geza and McCray [27] before calibration and by Wang and Melesse [26] for low flow predictions. Opposite results were reported by Geza and McCray [27] after calibration and by Wang and Melesse [26] for intermediate flows. Finally, no significant difference in performance of both STATSGO and SSURGO were found by Mukundan, et al. [28], and for high flows by Wang and Melesse [26].

In other studies, evaluating the detail needed from soil inventories, Boluwade and Madramootoo [30] showed taking more soil measurements does not necessarily increase the accuracy of the model predictions while Chaplot [24] showed that using fine-resolution soil information can improve streamflow prediction. Finally, Bossa, et al. [31] showed that including aggregated soil layers improved lateral flow predictions.

None of the above studies were carried out in developing countries that have a monsoon climate where rainfall in 4 months can exceed annual rainfall in temperate climate or where agricultural lands are often on the steep lands and used for subsistence farming. These countries often do not have the resources to use the detailed commercial soil data basis routinely because of the cost.

The objective of this research is, therefore, to investigate the impact of soil representation on streamflow prediction for watersheds greatly different in size that are in climates and landscapes that have not been studied before and where only digital global or continental soil inventories are easily available without cost.

The study areas selected were two watersheds in the upper reaches of the Blue Nile near Lake Tana in the Ethiopian highlands consisting of the 1316 km$^2$ Rib watershed and the 3.59 km$^2$ Gomit. Three soil data surveys were available for these watersheds consists of the Amhara Design and Supervision Works Enterprise (ADSWE), the Africa Soil Information System (AfSIS) and the Food and Agriculture Organization (FAO) soil inventories. The Soil and Water Assessment Tool (SWAT) was selected for simulating discharge because it is widely used in the developing world.

This research will aid watershed planners and modelers in deciding whether obtaining detailed soil inventory information is warranted by the improved accuracy of the output.

## 2. Materials and Methods

### 2.1. Study Area

The Rib watershed which is one of the four major tributary rivers in the eastern part the Lake Tana basin, north-western Ethiopian highlands. The watershed has a catchment area of 1316 km$^2$ and is located between 11°42′20″ to 12°11′11″ N and 37°42′43″ to 38°14′20″ (Figure 1). The Rib River starts on Mount Guna. The longest flow path to the gauging station is 118 km. The elevation ranges from 1790 to 4109 m. Seventy percent of the gauged watershed has gentle to steep slopes. The remaining part is the Fogera plain with slope of less than 1%. The climate is sub-humid with a mean annual rainfall of 1270 mm a$^{-1}$ (measured in the period from 1994 to 2013). Eighty percent of the annual rainfall occurs during the rainy season, which is between June to September. The annual potential evapotranspiration is 1428 mm a$^{-1}$ (millimeters per year) [32]. The agricultural system is predominantly rainfed and mixed crop-livestock production [33]. The primary crops cultivated is maize, barley, tef, wheat, beans, rice, and potato.

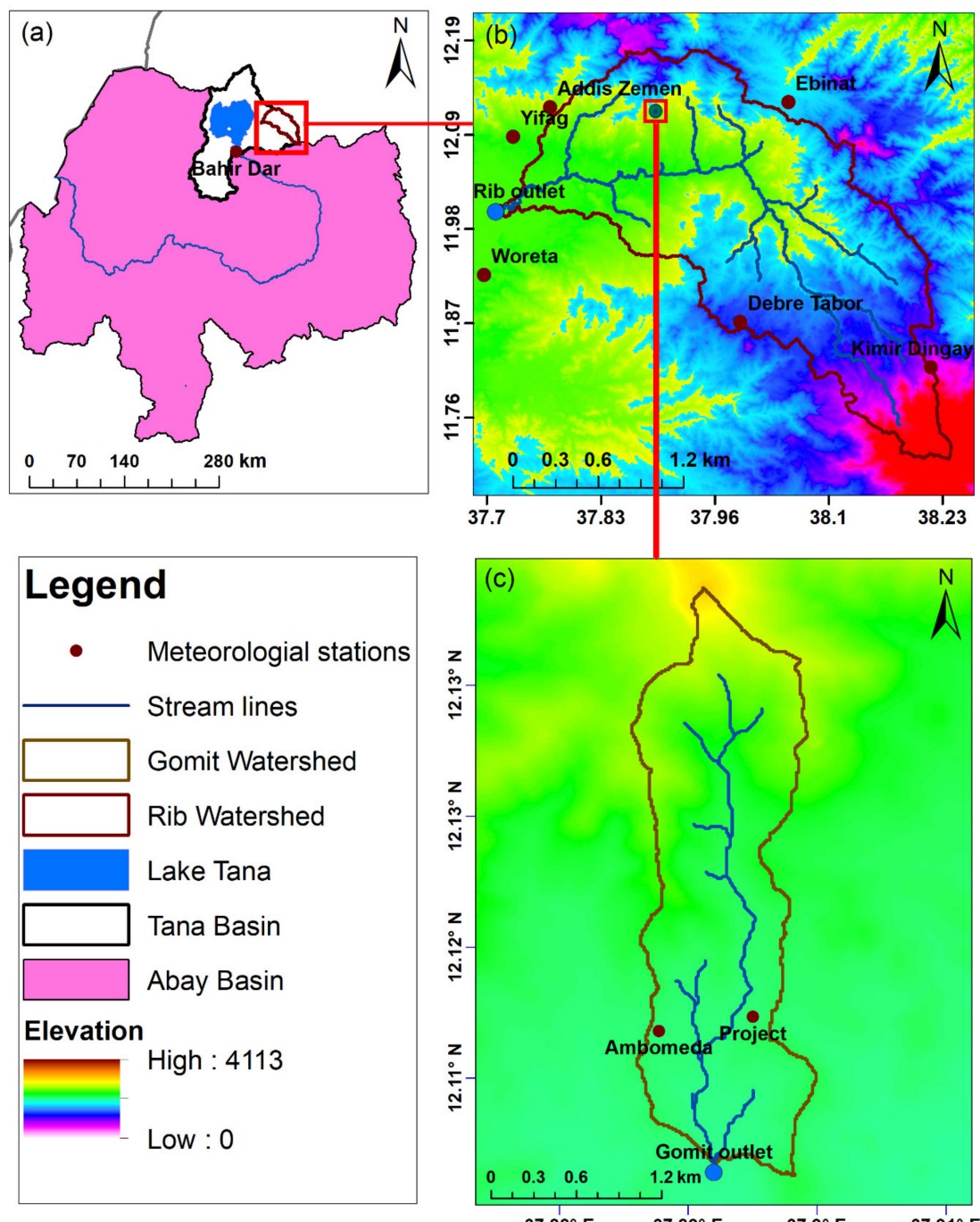

**Figure 1.** Location of the Rib and the Gomit watersheds and meteorological stations: (**a**) Upper Blue Nile basin showing the Lake Tana basin and the Rib watershed, (**b**) the Rib watershed showing the point location of the Gomit watershed, and (**c**) the Gomit watershed. The background on (**b**) and (**c**) is a digital elevation model (DEM).

The 3.59 km$^2$ Gomit watershed is nested in the upper Rib watershed between 12°6′9″ to 12°8′23″ N and 37°53′16″ to 37°54′ E. (Figure 1). The elevation range is between 1974 and 2612 m. The long-term mean annual rainfall (1997–2015) is 1265 mm a$^{-1}$ and potential evaporation (2005–2015) is 1428 mm a$^{-1}$ [32]. Twenty two percent of the watershed is cultivated. Crops are finger millet, tef, bean, maize, and niger seed. The remainder is bush, shrub and forest land, permanent grassland and various other minor land uses. Luvisols and leptosols, which are texturally clay and clay loam, are the dominant

soils in the watershed. Geologically the watershed is located in the region with alkaline to transitional tertiary basalts that is heavily faulted. More than half of the Gomit watershed has slopes greater than 30%. Land and water management (LWM) interventions were implemented starting in 2006. The LWM interventions consisted of contour soil bunds with 50 cm-deep infiltration furrows, waterways, micro basins, check dams, land protected from cattle entry and reforestation. The majority of the LWM practices are still visible.

*2.2. Data*

2.2.1. Hydrometeorological Data

Meteorological data for the period 1994-2013 was collected from the Ethiopian National Meteorological Agency (NMA). The meteorological stations are located at Addis Zemen, Debre Tabor, Ebinat, Kimir Dingay, Yifag, and Ambo Meda (Figure 1). Gomit watershed has its own rain gauge station to collect daily and event rainfall for the period 2015–2017. The meteorological data included rainfall, temperature for all stations and relative humidity, and wind speed for some of the stations. Most stations had only rainfall and temperature data. The Bahir Dar station data had the most complete data including hourly rainfall data and solar radiation and was used to calibrate the weather generator [31] Missing data was filled in using the calibrated weather generator.

Daily observed streamflow for the Rib river gauging station near Addis Zemen town (Figure 1) was obtained from the Ethiopian Ministry of Water, Irrigation and Energy (MoWIE) for the period 1994 to 2013. The daily streamflow was aggregated to monthly and used for sensitivity analysis, calibration and validation. We collected the stage readings for the Gomit watershed from July 2015 to December 2017 using masonry-modified rectangular weir at the outlet of the watershed (Figure 1). The stage-discharge curve for the Gomit watershed was determined by measuring the surface velocity with a float and multiplying it by two thirds to obtain the average stream velocity. The product of average velocity and wetted cross-sectional area is the discharge.

2.2.2. Spatial Data

The digital elevation model (DEM) with a spatial resolution of 12.5 m was used to discretize the watershed and create stream networks [34–36] was obtained from the Alaska Satellite Facility [37]. The DEM was also used to derive the slope.

The soil data were obtained from the Amhara Design and Supervision Works Enterprise (ADSWE), the African Soil Information Service (AfSIS) (http://africasoils.net/) and the Food and Agricultural Organization (FAO) of the United Nations (http://www.fao.org/soils-portal/soil-survey/soil-maps-and-databases/harmonized-world-soil-database-v12/en/). The ADSWE collected soil spatial information in 2014 for the Lake Tana basin using detailed field data survey which focused on soil morphological and physical characteristics. During the ADSWE field campaign, a total of 45,426 soil auger sites and 537 profiles were described and 1509 samples were collected [38]. One auger point represented 15 hm$^2$ (300 m × 500 m), and soil samples were taken from 845 pits. The soil data was collected up to a depth of 2 m; however, in areas that are dominated by rock, stones and/or hard soil layer, the depth was less than 2 m. Data were collected from up to six points in the soil profile to determine the soil chemical and physical properties. Finally, the ADSWE soil map was prepared at 1: 20,000 scale with 101 soil mapping units. Spatial representation of soil properties was weighted for each soil mapping unit.

The AfSIS soil inventory is available at 1:25,000 scale with six soil layers for the entire Africa [39]. This inventory integrates the Africa soil profiles legacy database with the AfSIS Sentinel Site (new soil samples) database to improve the spatial predictions of existing inventories. The compiled AfSIS data consists of 28 thousand sampling locations [39]. According to Hengl et al [39], two frameworks were implemented to producing spatial predictions of soil properties. In the first framework, soil properties were downscaled and predicted at 250 m resolution from previously mapped models at 1 km resolution using global soil prediction models. Models were developed using global soil classes in

the second framework. The FAO soil inventory covers the entire world and recognizes over 16,000 different soil mapping units at 1:1000,000 scale [40]. As Nachtergaele et al. [40], four source databases were used to compile the FAO soil inventory: (i) the European Soil Database (ESDB), (ii) the 1:1 million soil map of China, (iii) various regional Soil and Terrain digital (SOTER) databases, and (iv) the Soil Map of the World. Linkage of the attribute data with the spatial display in terms of soil units and the characterization of selected soil parameters, used to standardize the structure. The FAO soil map for the Upper Blue Nile basin, where the study watersheds are located, has three soil layers.

All the three soil inventories indicated that the Rib and the Gomit watersheds consisted mainly, of clay and clay loam soils (Table 1, Figure 2).

**Table 1.** Soil textural class of the Rib and Gomit watersheds for the Amhara Design and Supervision Works Enterprise (ADSWE), the African Soil Information Service (AfSIS), and the Food and Agricultural Organization (FAO) soil inventories.

| Watershed | Soil Inventory | Soil Textural Class | | |
| --- | --- | --- | --- | --- |
| | | Clay % | Clay Loam % | Others % |
| Rib | ADSWE | 53 | 39 | 8 |
| | AfSIS | 76 | 24 | - |
| | FAO | 63.4 | 36.5 | 0.1 |
| Gomit | ADSWE | 66 | 34 | - |
| | AfSIS | 40 | 60 | - |
| | FAO | 45 | 55 | - |

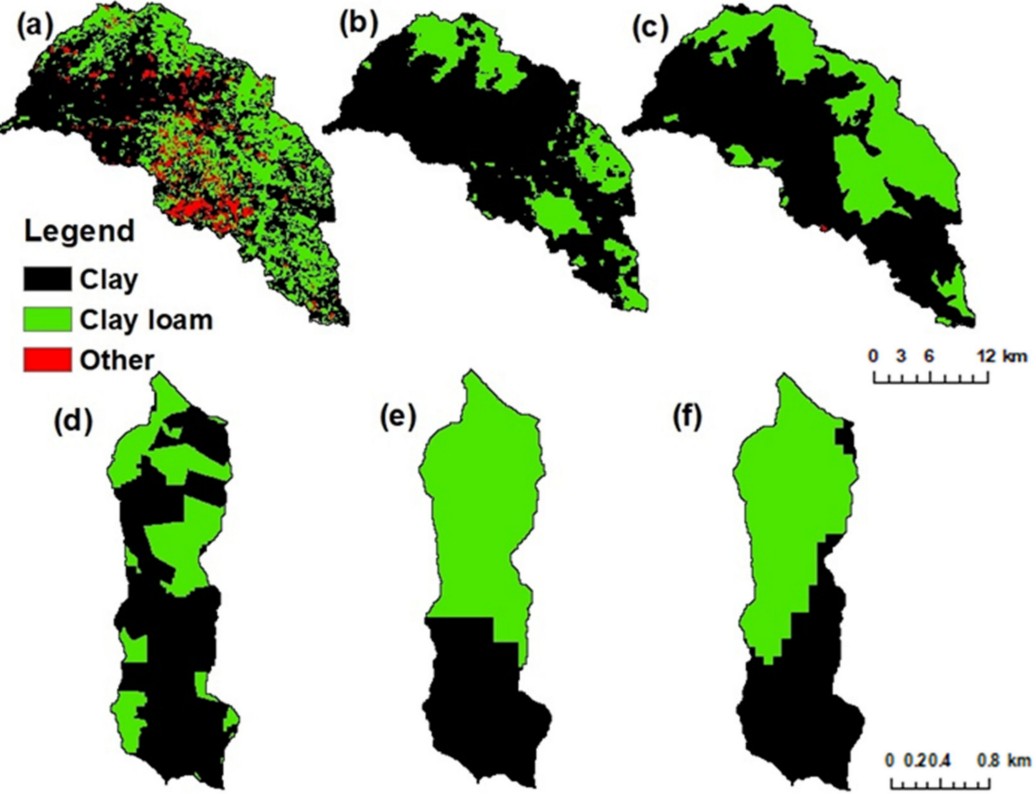

**Figure 2.** Spatial representation of the soil inventories of (**a–c**) the Rib and (**d–f**) Gomit watersheds; (**a,d**) the Amhara Design and Supervision Works Enterprise (ADSWE), (**b,e**) the African Soil Information Service (AfSIS) and (**c,f**) the Food and Agricultural Organization (FAO).

Only the ADSWE inventory showed that a significant amount of other soils existed in the watershed, namely 8% of the Rib. The soils consisted of loam, sandy loam, sandy clay and sandy clay loam (Table 1, Figure 2). The percentage of clay and clay loam soils varied among the three soils and was not consistent between the watersheds (Table 1). For example, for the Rib watershed the AfSIS soil inventory reported the lowest percentage of clay loam soils among the three soil inventories The opposite was the case for the Gomit watershed where the AfSIS soil inventory indicated the greatest portion of clay (Table 1, Figure 2).

The land-use data for the Rib watershed was extracted from the Upper Blue Nile (Abay) River master plan study, which was obtained from the Ethiopian Ministry of Water, Irrigation and Energy [41]. The land-cover map for Rib watershed was predominantly (89%) cultivated land, and the remaining part of the watershed was grassland, afro-alpine vegetation and urban (Figure 3a). The land-cover map of the Gomit watershed was developed using Google Earth image in combination with field observations (Figure 3b). Object-based image classification using multiresolution segmentation classification algorithm was applied to develop the Gomit watershed land cover. This classification approach has been proven to provide better classification results than per-pixel classification approaches in vegetation classification, especially for fine spatial resolution data [42–44]. Field investigation was used to verify the land cover types of the watershed. For the Gomit, 57 % of the watershed was comprised of bush and shrub land. Cultivated land covered 22 % of the watershed and 7 % consisted of forested land. The remaining part of the watershed (14 %) consisted of a farm village, grassland, plantations, and settlements.

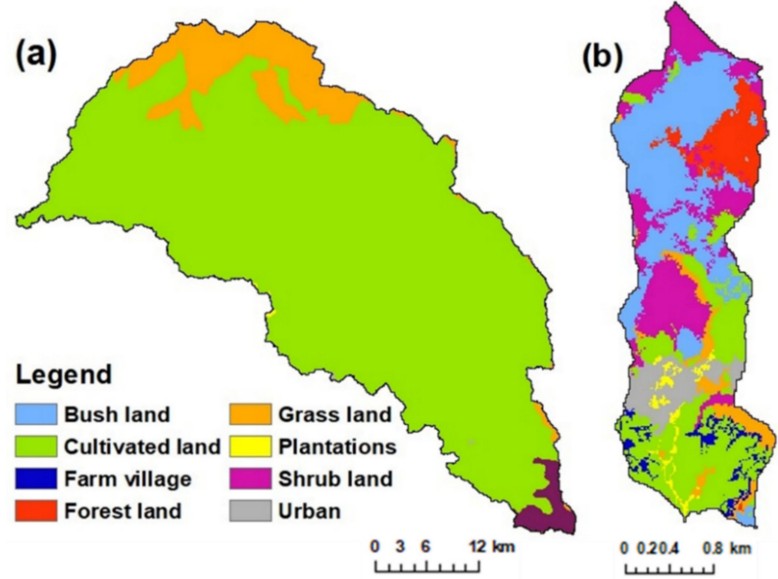

**Figure 3.** Landcover maps of (**a**) the Rib and (**b**) Gomit watersheds.

*2.3. Analysis*

2.3.1. Soil and Water Assessment Tool (SWAT) Model Description

In this study, soil and water assessment tool (SWAT) hydrological model was used because it is widely used physically distributed model in the globally designed to predict the impact of land-use, climate and agricultural management changes on water, sediment, and agricultural chemical yields in gauged and ungauged watersheds [45]. In addition, SWAT uses spatially distributed data on topography, soils, land cover, land management, and weather compared with other hydrological models like Hydrologiska Byråns Vattenbalansavedlning (HBV) and Hydrologic Engineering Center - Hydrologic Modeling System (HEC-HMS) [46]. It operates on a daily time step. The model has a platform that permits the user to partition the watershed into sub-basins, sub-basins to hydrologic

response units (HRUs). Runoff curve number method determines the amount of runoff in each HRU. SWAT has land and routing phases in simulating the hydrology of a specific watershed [34]. The land phase controls the amount of water, sediment, nutrient and pesticide loadings to the main channel from each sub-basin. The routing phase determines the movement of water, sediment and other pollutants in the channel network to the watershed outlet [34]. The land phase of the hydrological cycle is simulated by SWAT using the water balance equation:

$$SW_t = SW_0 + \sum_{i=1}^{t} \left( R_{day} - Q_{surf} - E_a - w_{seep} - Q_{gw} \right) \tag{1}$$

where $SW_t$ is the final soil water content (mm), $SW_0$ is the initial soil water content on day $i$ (mm), $t$ is the time (days), $R_{day}$ is the amount of precipitation on day i (mm), $Q_{surf}$ is the amount of surface runoff on day $i$ (mm), $E_a$ is the amount of evapotranspiration on day i (mm), $w_{seep}$ is the amount of water entering the vadose zone from the soil profile on day $i$ (mm), and $Q_{gw}$ is the amount of return flow on day $i$ (mm) [47].

2.3.2. Model Setup and Calibration

The steps in running SWAT include defining watershed and stream network, hydrological response unit (HRU) analysis, sensitivity analysis, calibration and validation using the three soil inventories (Figure 4). The discretization for the Rib watershed was created in SWAT with an area threshold of 2500 hm$^2$. The threshold in the Gomit watershed was 10 hm$^2$.

To evaluate the model response for the different soil representations, the model should have a sufficient number of sub-basins. We did not set equal threshold areas for both of the watersheds (Figure 4a), because the required threshold area for the small watershed creates too many sub-basins for the large watershed that would need smart computer architecture [48]. The digitized stream networks were used to improve hydrographic segmentation and sub-watershed boundary delineation. The watershed discretization provided 35 sub-basins for the Rib watershed and 19 for the Gomit watershed (Figure 4a). The HRUs were defined considering all land use, soil and slope data (i.e., using zero percent as a threshold to eliminate smaller units, Figure 4b). Before using the three soils for the HRU definition, the soils map was resampled to 12.5 × 12.5 m using nearest neighbor resampling technique to create similar resolution as to the DEM data [49–51]. Similar to the soil map, the land-cover map was resampled to create similar resolution to the DEM data. The number of HRUs for the Rib watershed for the ADSWE soil inventory were 2617, for the AfSIS inventory were 6123 and 374 for the FAO inventory, and for the Gomit watershed, 430, 341 and 248, respectively (Figure 4b).

The Rib and Gomit SWAT models were calibrated using the Sequential Uncertainty Fitting Version-2 (SUFI-2) optimization algorithm included in the SWAT Calibration and Uncertainty Program (SWAT-CUP) [52]. The SUFI-2 algorithm was selected because of its satisfactory performance in the study watershed [53]. The model calibration was performed after selecting the sensitive streamflow parameters [30] (Figure 4c). Acceptable ranges of flow parameter and the type of change to be applied to the parameter were collected from literature and SWAT-CUP absolute SWAT values (see Table S1 in the Supplementary Materials). A similar set of model parameters was used in both watersheds models to ensure a reasonable comparison between the watersheds (Table 4). The calibration was conducted by considering and excluding sensitive soil parameters to evaluate the impact of the different soil inventories in streamflow prediction (Figure 4d). The Rib watershed was calibrated using observed streamflow for the period 1996 to 2007 and validated for the period 2008 to 2013 (Figure 4e). The observed streamflow data for the period 1994 to 1995 were used for model warm up. For the Gomit watershed, observed streamflow data were available only for the period of July 2015 to December 2017, which was used for model calibration. The data for 2014 were used to warm-up the model simulation with rainfall of the Ambomeda station. Since the model validation was short, all the observed data for the Gomit were used for calibration, and model validation was not conducted.

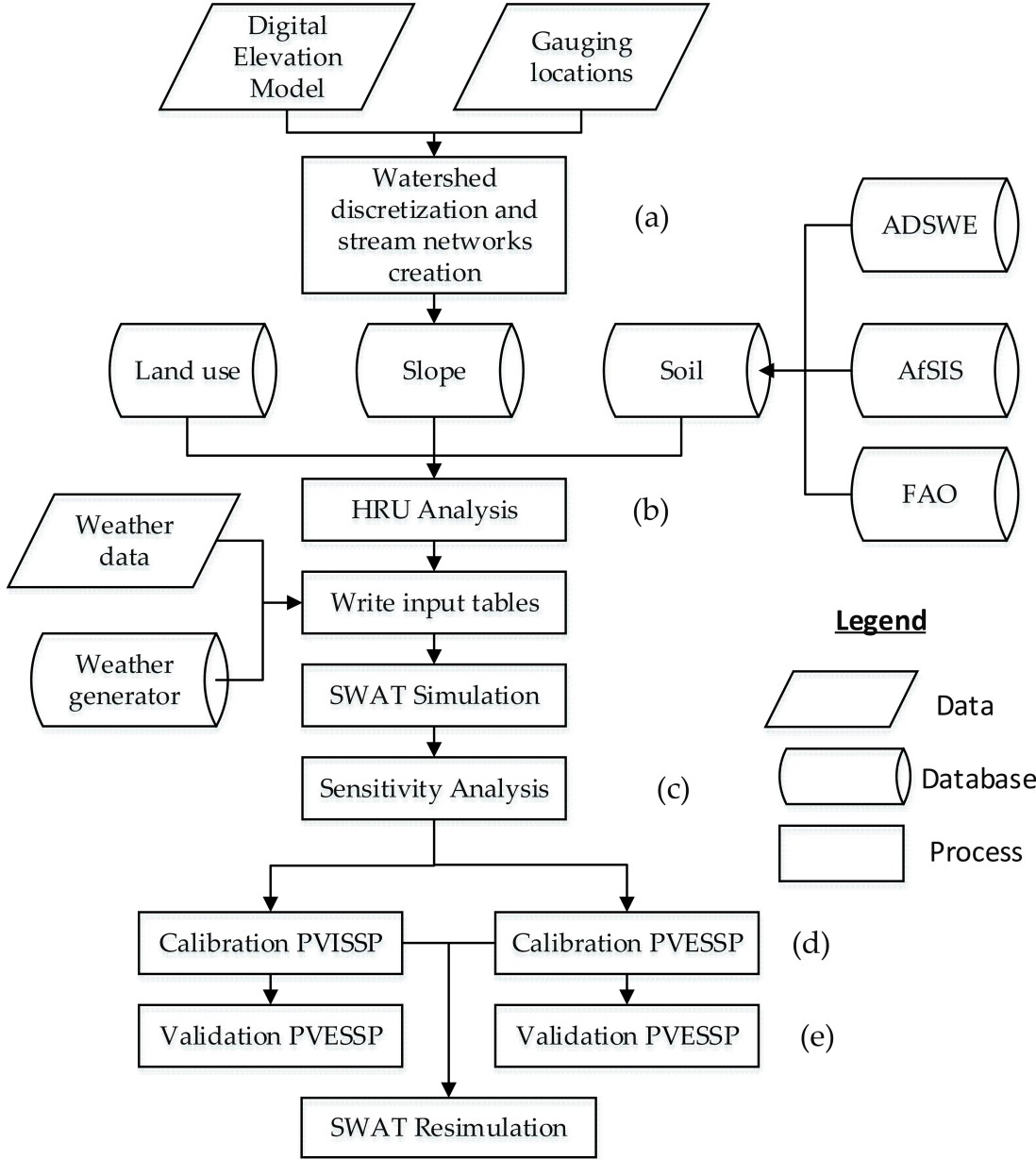

**Figure 4.** Flow chart showing the Soil and Water Assessment Tool (SWAT) hydrological modeling procedures of this study using three soil inventories: Amhara Design and Supervision Works Enterprise (ADSWE), Africa Soil Information Service (AfSIS) and Food and Agricultural Organization (FAO). The letters in parenthesis represents the various steps in preparation of the input data and running of the model (**a**) watershed discretization; (**b**) data input and Hydraulic Response Unit (HRU) selection; (**c**) sensitivity analysis (**d**) model calibration and (**e**) model validation. PVISSP represents calibration including the soil parameters, and PVESSP stands for calibration that did not include the soil parameters.

### 2.3.3. Model Performance Evaluation

The effect of soil representation on streamflow prediction was evaluated before and after model calibration using descriptive statistics, goodness-of-fit criteria and paired samples t-test [54–56]. The descriptive statistics used for evaluation were maximum, minimum and mean. The goodness-of-fit evaluation methods considered for the analysis were coefficient of determination ($R^2$), Nash–Sutcliffe efficiency (NSE), and relative volume error (RVE).

The Nash–Sutcliffe efficiency (NSE) is a normalized statistic that determines the relative magnitude of the residual variance compared to the measured data variance [57]. NSE indicates how well the plot of observed versus simulated data fits the 1:1 line. NSE is computed as:

$$\text{NSE} = 1 - \frac{\sum_{i=1}^{n} (O_i - P_i)^2}{\sum_{i=1}^{n} (O_i - \bar{O})^2} \tag{2}$$

where $O$ is observed, $P$ is simulated values and the over bar denotes the mean for the entire time period of the evaluation.

NSE values can generally range between $-\infty$ and 1; an NSE value of 1 represents a perfect match between simulated and observed data. NSE value between 0 and 1 are generally considered as acceptable levels of model performance, whereas values <0 indicates that the mean observed value is a better predictor than the simulated value, which indicates an unacceptable performance. Generally, a model performance is considered satisfactory when the NSE value is more than 0.5 [55].

The coefficient of determination ($R^2$) explains the fraction of the total variance and ranges from 0 to 1. $R^2$ with a value of one indicates excellent agreement, and the value of zero reflect that there is no co-relation between the simulated and observed values [58]. $R^2$ is defined as:

$$R^2 = \left( \frac{\sum_{i=1}^{n} \left(O_i - \bar{O}\right)\left(P_i - \bar{P}\right)}{\sqrt{\sum_{i=1}^{n} (O_i - \bar{O})^2} \sqrt{\sum_{i=1}^{n} (P_i - \bar{P})^2}} \right)^2 \tag{3}$$

The relative volume error ($RV_E$) is used to evaluate the volumetric difference between simulated and observed streamflow, which can vary between $+\infty$ and $-\infty$. The method is used in combination with another objective function that evaluates the overall shape agreement between the two variables. An RVE value of zero indicates there is no volumetric difference between simulated and observed streamflow. Generally, a relative volume error between ±25% indicates a satisfactory model performance [55]. The equation that is used to compute $RV_E$ is:

$$RV_E = \left( \frac{\sum_{i=1}^{n} P_i - \sum_{i=1}^{n} O_i}{\sum_{i=1}^{n} O_i} \right) \tag{4}$$

An independent two sample *t*-test, which compares the means of two unrelated groups of samples [59], was used to evaluate the similarity of model simulations. The difference between simulations was considered significant when the p-value was less than 5%.

## 3. Results

### 3.1. Comparing Model Components before Calibration

Mean annual water balance simulated before model calibration for the Rib (1996–2007) and the Gomit (2015–2017) watersheds using each of the three soil inventories is shown in Table 2. There was a distinct difference between the water balance components for each of the simulations. For example, the surface runoff with the ADSWE inventory was the lower in the Rib watershed compared to the surface runoff predicted with the AfSIS and the FAO inventories. The opposite was true for the Gomit watershed where simulated surface runoff was greater with the ADSWE inventory. Simulation with the AfSIS and the FAO inventories indicated that approximately two thirds of the rainfall in the Rib and one sixth in the Gomit watershed was lost either as overland flow or baseflow (Table 2). Simulations with the ADSWE inventory in the Gomit watershed resulted in the lowest amount evaporation and the greatest amount of surface runoff (Table 2).

**Table 2.** Mean SWAT-simulated annual water balance components (in mm a$^{-1}$) using the soil inventories of the Amhara Design and Supervision Works Enterprise (ADSWE), the Africa Soil Information Service (AfSIS) and the Food and Agricultural Organization (FAO) for the Rib watershed (1996–2007) and the Gomit watershed (2015–2017) before model calibration.

| Watershed | Soil in-Ventories | Rainfall | Evapotranspiration | Surface Runoff | Baseflow | Recharge to Deep aquifer | Re-evap shallow Aquifer |
|---|---|---|---|---|---|---|---|
| | ADSWE | 1276 | 466 | 340 | 426 | 20 | 24 |
| Rib | AfSIS | 1276 | 525 | 399 | 314 | 14 | 24 |
| | FAO | 1276 | 465 | 401 | 368 | 19 | 24 |
| | ADSWE | 988 | 640 | 153 | 128 | 5 | 73 |
| Gomit | AfSIS | 988 | 836 | 41 | 103 | 0 | 62 |
| | FAO | 988 | 760 | 117 | 60 | 4 | 70 |

In both watersheds, the simulated discharge (both surface runoff and baseflow) overestimated the observed discharge. For the Rib watershed, the simulated mean annual (1996 to 2013) streamflow ranged from 713 mm to 769 mm a$^{-1}$ and were not statistically different (Table 2). The long-term mean annual observed streamflow of 370 mm a$^{-1}$ which was significantly less ($p < 0.5$). Thus, there was a two-fold overestimation of the observed streamflow for the three soil inventories (Figures 4a and 5a).

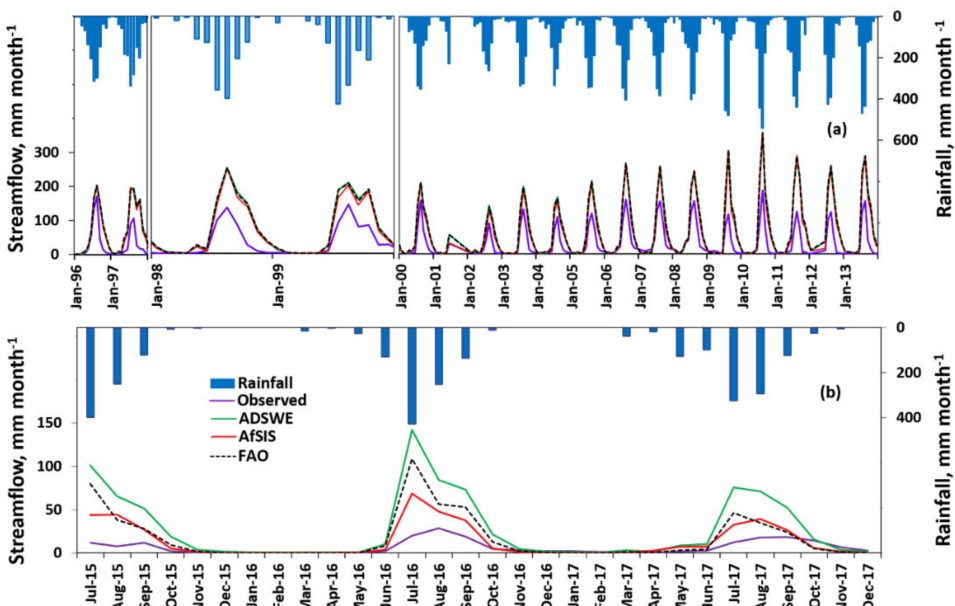

**Figure 5.** Monthly streamflow simulation before calibration using the soil inventories of the Amhara Design and Supervision Works Enterprise (ADSWE), the Africa Soil Information Service (AfSIS) and the Food and Agricultural Organization (FAO). To display the detail, the timescale for the 1999 and 2000 was extended; (**a**) the Rib watershed and (**b**) the Gomit watershed.

In the Gomit watershed, the average simulated watershed outflow over the three years was 200 mm a$^{-1}$ or around 20% of the rainfall (Table 2) and was significantly greater than the observed three year average of 65 mm (7% of the precipitation). Consequently, the flows during the rainfall events were severely overpredicted (Figure 5b, Figure 6b). Adem et al. [60] explained that discharge at the outlet was small because subsurface flow through the faults was significant. The faults transported water to a different basin. The soil inventories failed to include the faulted geology of the watershed and hence the model directed most rainfall in excess of evaporation to the outlet (Table 2).

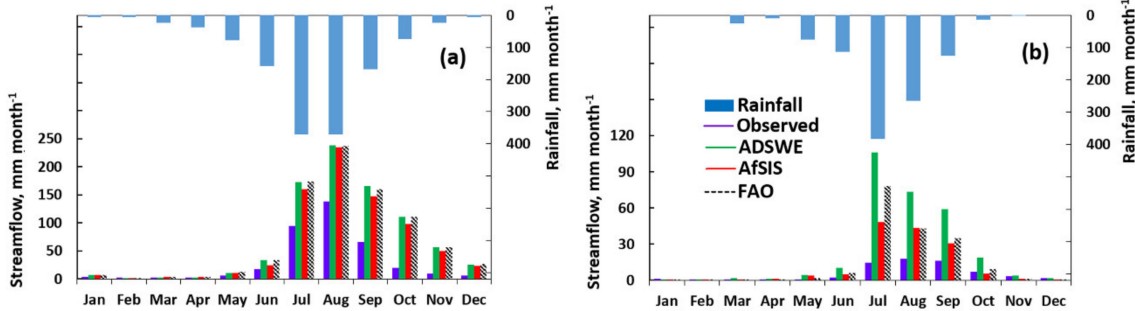

**Figure 6.** Mean monthly streamflow before calibration using the soil inventories of the Amhara Design and Supervision Works Enterprise (ADSWE), the Africa Soil Information Service (AfSIS) and the Food and Agricultural Organization (FAO). (**a**) The Rib watershed and (**b**) the Gomit watershed.

Statistically, the negative Nash–Sutcliffe coefficients in Table 3 indicates that the average streamflow was a better predictor for discharge than the discharge predicted by SWAT. Similarly, the $RV_E$ >100% showed that model performance was poor in both watersheds for all three soil inventories (Table 3). Finally, the coefficient of determination ($R^2$) for both watersheds was greater than 0.6, indicating that both simulated and observed runoff were greater during the rain phase than in the dry monsoon phase (Figures 5b and 6b) and it does not relate to satisfactory simulations because only the slope close to 1 would indicate the amounts were nearly the same (Table 3).

**Table 3.** Goodness-of-fit evaluation for SWAT model streamflow prediction of the Rib and Gomit watersheds using the Amhara Design and Supervision Works Enterprise (ADSWE), the Africa Soil Information Service (AfSIS) and the Food and Agricultural Organization (FAO) soil inventories. Values refer for the streamflow simulations before model calibration. $R^2$ is the coefficient of determination, NSE is the Nash–Sutcliffe efficiency and $RV_E$ is relative volume error.

|  | ADSWE | | AfSIS | | FAO | |
|---|---|---|---|---|---|---|
|  | **Rib** | **Gomit** | **Rib** | **Gomit** | **Rib** | **Gomit** |
| $R^2$ | 0.83 | 0.70 | 0.85 | 0.72 | 0.83 | 0.6 |
| NSE | −0.77 | −22.5 | −0.41 | −2.7 | −0.73 | -8.5 |
| $RV_E$, % | 125 | 326 | 108 | 110 | 126 | 166 |

### 3.2. Sensitivity Analysis

In the sensitivity analysis, 14 input parameters were selected out of 23. To ensure a reasonable comparison between the watersheds, sensitive parameters (e.g. GW_REVAP) in one of the study watershed were included in the other watershed (Table 4). The four soil-related parameters consisted of the soil available water content (SOL_AWC), soil albedo (SOL_ALB,) soil depth (SOL_Z), and soil hydraulic conductivity (SOL_K). Table 4 ranks the sensitivity of the 14 input parameters. The ranking is similar for the three inventories for each watershed but dissimilar between the two watersheds.

The four most sensitive input parameters that provide the largest change in discharge for the same relative change in the input value in the Rib watershed are groundwater delay (GW_DELAY), groundwater "revap" coefficient (GW_REVAP), SCS runoff curve number (CN2), and threshold depth shallow aquifer for return flow (GWQMN) (Table 4).

**Table 4.** Sensitivity analysis of fourteen input parameters for simulating streamflow in the Rib and Gomit watersheds using the Amhara Design and Supervision Works Enterprise (ADSWE), the Africa Soil Information Service (AfSIS) and the Food and Agricultural Organization (FAO) soil inventories.

| Parameter | Description | Sensitivity Ranking | | | | | |
|---|---|---|---|---|---|---|---|
| | | Rib | | | Gomit | | |
| | | ADSWE | AfSIS | FAO | ADSWE | AfSIS | FAO |
| **GW_DELAY** | Groundwater delay (days) | 1 | 2 | 1 | 9 | 11 | 5 |
| GW_REVAP | Groundwater "revap" coef. | 2 | 3 | 3 | 18 | 16 | 15 |
| CN2 | SCS runoff curve number | 3 | 1 | 2 | 1 | 1 | 1 |
| GWQMN | Threshold depth shallow aquifer for return flow | 4 | 4 | 4 | 12 | 12 | 8 |
| RCHRG_DP | Deep aquif percolation frac | 5 | 5 | 5 | 5 | 10 | 4 |
| ESCO | soil evaporation comp fac. | 6 | 7 | 6 | 2 | 2 | 2 |
| ALPHA_BNK | Baseflow fac. for bank sto. | 7 | 8 | 7 | 7 | 8 | 7 |
| SOL_AWC | Avail. water cap layer | 8 | 11 | 10 | 3 | 3 | 3 |
| ALPH_BF_D | Baseflow fac. deep aquifer | 9 | 10 | 9 | 22 | 22 | 17 |
| CANMX | Maximum canopy storage, | 10 | 6 | 8 | 4 | 5 | 6 |
| SOL_Z | Depth surf. to bottom layer | 18 | 9 | 13 | 11 | 6 | 10 |
| SOL_K | Saturated hydraulic cond. | 19 | 20 | 20 | 10 | 4 | 11 |
| SOL_ALB | Moist soil albedo | 20 | 18 | 17 | 6 | 9 | 9 |
| BIOMIX | Biological mixing eff. | 21 | 22 | 23 | 8 | 7 | 12 |

## 3.3. Comparing Model Components for the Period of Calibration

### 3.3.1. General Observations

The water balance components changed considerably after model calibration in both watersheds. Calibrated parameters, their range and fitted values for the streamflow simulations using the ADSWE, the AfSIS and the FAO in the Rib and Gomit watersheds are shown in Table S1 in the Supplemental Material. In both watersheds, after the model was calibrated the simulated streamflow (consisting of both surface runoff and baseflow) was reduced, and the evapotranspiration and recharge to the deep aquifer were increased (compare Tables 2 and 5). In the Rib watershed, the evaporation (both evapotranspiration from the plants and the evaporation from the shallow groundwater) increased modestly and accounted for between 53% and 59% of the rainfall depending on the digital soil inventory used. In addition, in the Rib watershed, the percolation to the deep groundwater increased from 1% to 12% with the AfSIS and the FAO inventories, and decreased from 2% to none with the ADSWE soil inventories (Table 5). The discharge at the outlet in the Gomit watershed was less due mainly to the increased evapotranspiration of crop, bush and forests. The total the Gomit watershed evaporation accounted for 89% to 97% of the rainfall.

### 3.3.2. The Rib Watershed

Despite the improved model performance in the Rib watershed during both calibration and validation of the SWAT model (with either of the three soil inventories and independent whether the soil parameters were considered in the calibration), the predicted discharge at the outlet was greater than the observed streamflow. Only in the first year of simulation was discharge underpredicted. In particular, the simulated discharge overpredicted the peaks in August and falling limbs of the hydrographs from September to the beginning of the rain phase in May and June (Figures 7 and 8). The overprediction was greater for the ADSWE and the FAO soil inventories compared with the AfSIS inventory.

In the Rib watershed, where the average mean annual observed streamflow over the period 1996–2007 was 370 mm $a^{-1}$, the mean annual simulated streamflow with the ADSWE, the AfSIS, and the FAO soil inventories ranged from 389 to 495 mm $a^{-1}$, respectively (Table 5). The *t*-test showed that the difference between the mean annual observed streamflow and simulated streamflow with the ADSWE and the AfSIS soil inventories were not statistically significant. However, the observed monthly streamflow and simulated monthly streamflow with the FAO soil inventory were significantly

different. Finally, the streamflow simulations with the three soil inventories independent whether soil parameters in the calibration were considered, were not significantly different.

**Table 5.** Mean simulated annual water balance components in mm a$^{-1}$ for the calibrated SWAT model using the soil inventories of the Amhara Design and Supervision Works Enterprise (ADSWE), the Africa Soil Information Service (AfSIS) and the Food and Agricultural Organization (FAO) for the Rib watershed (calibration period, 1996–2007) and the Gomit watershed (2015–2017) after model calibration. PVISSP represent values of water balance components for the simulations with all calibrated parameters including the soil parameters, and PVESSP represent values of water balance components for simulations that did not include the soil parameters during calibration.

| Water-shed | | Soil Data | Rainfall | Evapo-transpiration | Surface Runoff | Base-Flow | Recharge to Deep Aquifer | Re-evaporation from Shallow Aquifer |
|---|---|---|---|---|---|---|---|---|
| Rib | PVISSP | ADSWE | 1276 | 537 | 323 | 172 | 0 | 213 |
| | | AfSIS | 1276 | 606 | 256 | 133 | 134 | 133 |
| | | FAO | 1276 | 534 | 327 | 91 | 159 | 146 |
| | PVESSP | ADSWE | 1276 | 536 | 318 | 176 | 0 | 215 |
| | | AfSIS | 1276 | 595 | 269 | 124 | 137 | 136 |
| | | FAO | 1276 | 529 | 335 | 85 | 135 | 192 |
| Gomit | PVISSP | ADSWE | 988 | 840 | 26 | 64 | 19 | 36 |
| | | AfSIS | 988 | 911 | 6 | 73 | 0 | 0 |
| | | FAO | 988 | 897 | 12 | 40 | 16 | 57 |
| | PVESSP | ADSWE | 988 | 855 | 11 | 68 | 0 | 52 |
| | | AfSIS | 988 | 897 | 5 | 86 | 0 | 0 |
| | | FAO | 988 | 894 | 15 | 40 | 24 | 17 |

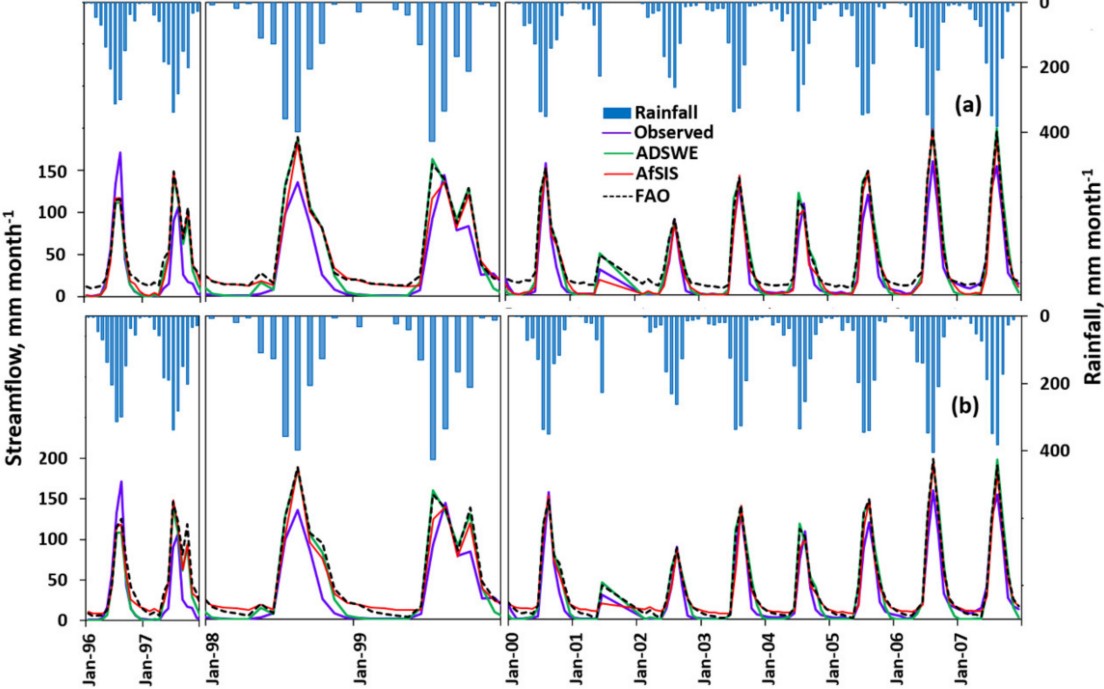

**Figure 7.** Monthly observed and simulated streamflow during calibration and validation using the soil inventories of the Amhara Design and Supervision Works Enterprise (ADSWE), the Africa Soil Information Service (AfSIS) and the Food and Agricultural Organization (FAO). Calibration period was from 1996 to 2007 and validation period from 2008 to 2013; the timescale for the 1999 and 2000 was extended to display the detail; (**a**) model calibration included the soil parameters, and (**b**) model calibration did not include the soil parameters.

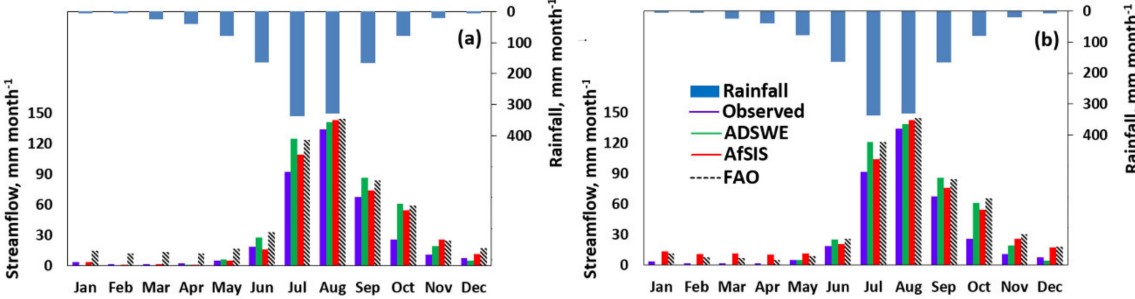

**Figure 8.** Long-term mean monthly observed and simulated discharge using the digital soil inventories of the Amhara Design and Supervision Works Enterprise (ADSWE), the Africa Soil Information Service (AfSIS) and the Food and Agricultural Organization (FAO); (**a**) model calibration included the soil parameters, and (**b**) model calibration did not include the soil.

### 3.3.3. The Gomit Watershed

In the Gomit watershed, where the average mean annual observed streamflow over the period 2015–2017 was 65 mm a$^{-1}$, the mean annual simulated streamflow with the ADSWE, the AfSIS, and the FAO soil inventories ranged from 52 to 91 mm a$^{-1}$, respectively (Table 5). While the rainfall was greatest in July, the peak of runoff tended to occur in August (Figure 9). Independently of the soil inventory used, the simulated runoff peak coincided with the month of highest rainfall. The annual runoff ratio varied between 9% for the ADSWE and the AfSIS inventories to 5% for the FAO inventory. According to the *t*-test, no significant differences were found between observed and none of the calibrated streamflow despite some of them did not include the soil parameters in the calibration procedure.

Unlike in the Rib watershed, the inclusion of the soil parameters (SOL_AWL, SOL_K, SOL_Z, and SOL_ALB) changed the values of the other parameters and the ranking of the sensitivities (Table 4), and improved the NSE values but not the RV$_E$ (Table 6) during calibration. Unfortunately, the observed record was too short for the validation. Thus, soil information was found to have a greater but not significant effect on the modelled hydrological processes in the smaller Gomit watershed when compared to the larger Rib watershed.

**Table 6.** Evaluation of streamflow simulation using the digital soil inventories of the Amhara Design and Supervision Works Enterprise (ADSWE), the Africa Soil Information Service (AfSIS) and the Food and Agricultural Organization (FAO) for the Rib and the Gomit watersheds. NSE and RVE refer to the Nash–Sutcliffe efficiency and relative volume error, respectively. PVISSP represents goodness-of-fit values for the simulations of all calibrated parameters including the soil parameters, and PVESSP represents goodness-of-fit values for simulations that did not include the soil parameters during calibration.

| Watershed | Process | *Criteria* | *ADSWE* | | *AfSIS* | | *FAO* | |
|---|---|---|---|---|---|---|---|---|
| | | | **PVISSP** | **PVESSP** | **PVISSP** | **PVESSP** | **PVISSP** | **PVESSP** |
| Rib | Calibration | NSE | 0.78 | 0.79 | 0.83 | 0.83 | 0.75 | 0.75 |
| | | RV$_E$ | 7 | 25 | 35 | 35 | 51 | 43 |
| | Validation | NSE | 0.41 | 0.46 | 0.57 | 0.58 | 0.44 | 0.43 |
| | | RV$_E$ | 53 | 47 | 54 | 55 | 68 | 69 |
| Gomit | Calibration | NSE | 0.63 | 0.60 | 0.59 | 0.46 | 0.67 | 0.67 |
| | | RV$_E$ | 14 | 1 | 12 | 22 | 9 | 1 |

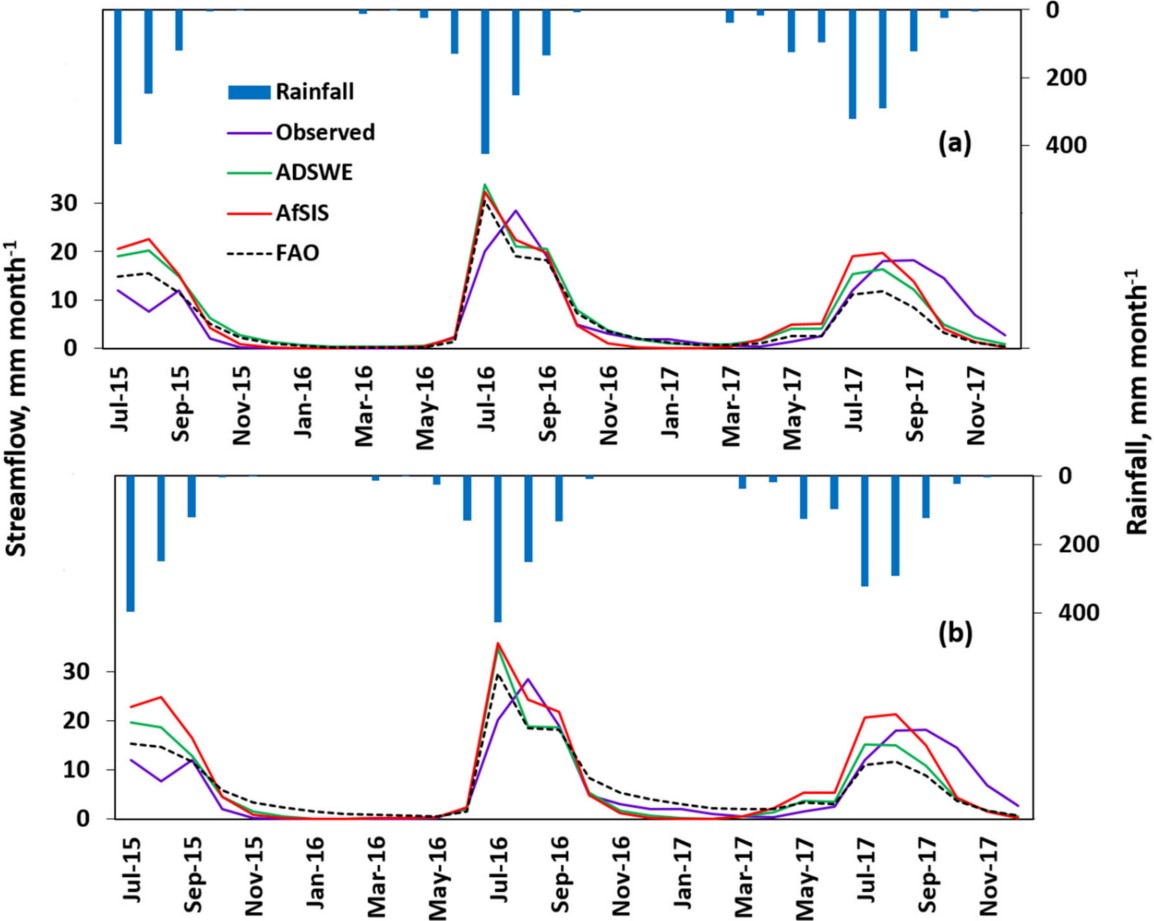

**Figure 9.** Mean monthly observed and simulated streamflow for the Gomit watershed using the digital soil inventories of the Amhara Design and Supervision Works Enterprise (ADSWE), the Africa Soil Information Service (AfSIS) and the Food and Agricultural Organization (FAO). (**a**) considering 14 sensitive parameters including four soil parameters during calibration and (**b**) without the inclusion of the soil parameters during calibration.

## 4. Discussion

The discharge was predicted in two watersheds using the SWAT model with three soil inventories with varying complexity. The soil inventories used were the ADSWE, the AfSIS and the FAO. The soil map of the two watersheds showed large spatial differences for the three soil inventories (Figure 2; Table 1). For example, the amount of clay loam varied from 34% in the ADSWE inventory to 60% in the AfSIS soil inventory in the Gomit watershed (Table 1).

Independently of the digital soil inventory used, the discharge at the outlet of the Rib and the Gomit watersheds were overpredicted before the models were calibrated (Table 2, Figures 5 and 6). After calibration the discharge in the Rib watershed was still overpredicted but not in the Gomit watershed (Table 5, Figures 7 and 8). The observed stream flows in the Rib and Gomit watersheds are less than those of the watersheds to south of Lake Tana, because of the presence of fractures and fault zones in the watersheds [60]. Dessie, et al. [61] reported that these geological formations cause regional subsurface flow, which reduce streamflow at the watershed outlets. In addition, the observed peak streamflow of the Rib river could be reduced due to overbank flooding [62,63]. However, the bank flooding cannot explain the deviation between observed and predicted discharge during the dry phase (Figures 7 and 8). Moreover, since the Gomit watershed streamflow, measured by us, was reliable, and nested within the Rib watershed, it supports the premise that the low runoff generation in the Rib

watershed was due to regional subsurface water loss in geological fault zone formations and not by measurement error.

The use of the three soil inventories did not affect the simulated discharge significantly because models runs with calibrated the soil parameters were not significantly different from those that did not include the soil parameters. There were some slight non-significant differences in performance of the soil inventories: the ADSWE soil inventory simulations provided in the most accurate in the Rib watershed discharge ($RV_E \leq 25\%$); the FAO soil inventory in the Gomit watershed predicted discharge best ($RV_E \leq 9\%$) (Table 6). The AfSIS soil inventory gave the best streamflow prediction before calibration in the in the Gomit watershed. (Tables 2 and 5).

Our modeling results on the effect of level soil inventories on streamflow prediction for two watersheds in a monsoon climates are consistent with other studies in temperate climates that the detail of the soil information was insignificant [21–23,26–30]. Thus, in regions with a monsoon climate and in temperate climates and under widely varying land-use conditions, the choice of the soil inventory affects the streamflow predictions only minimally.

## 5. Conclusions

In this study, we used digital soil inventories from the Amhara Design and Supervision Works Enterprise (ADSWE), the African Soil Information Service (AfSIS), and the Food and Agricultural Organization (FAO) to evaluate the effect of soil data representation and spatial variability on simulating the hydrology of the watershed. The evaluation was conducted using the SWAT hydrological model. This model is widely used in developing countries because of its extensive support structure.

The findings are that in the highlands of Ethiopia with a monsoon climate the details of soil spatial information do not affect discharge predictions significantly and are in agreement with those in temperate climate and other less steep landscapes [22–30]. The additional information provided in more complex soil inventories can change the fitted parameters in the model, but do not affect the flow at the outlet of the watershed.

**Supplementary Materials:** The following are available online at http://www.mdpi.com/2306-5338/7/1/8/s1: Table S1: Calibrated parameters, their range and fitted values for the streamflow simulations using the ADSWE, the AfSIS and the FAO in the Rib and the Gomit watersheds

**Author Contributions:** All authors were involved in the conceptualization and methodology. A.A.A. collected the data, made the model runs, analyzed the data and wrote the first draft. All authors were involved in writing the final version of the manuscript. All authors have read and agreed to the published version of the manuscript.

**Funding:** This publication was made possible through support provided by CGIAR Research Program on Water, Land and Ecosystem's East Africa focal regional program and by the Feed the Future Innovation Lab for Small Scale Irrigation through the U.S. Agency for International Development, under the terms of Contract No. AID-OAA-A-13-0005. The contents of the paper do not necessarily reflect the views of CGIAR and USAID.

**Acknowledgments:** Bahir Dar Institute of Technology and the Postgraduate School at Bahir Dar provided financial support during the write-up.

**Conflicts of Interest:** The authors declare that they do not have conflicts of interest.

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
