# Peer review of "Assessing Digital Soil Inventories for Predicting Streamflow in the Headwaters of the Blue Nile"

_hydrology, doi:10.3390/hydrology7010008_

Round 1
Reviewer 1 Report
Dear Sirs,
This is an interesting, practical, and valuable paper concerning water resources assessment for developping countries. The methodological approach is well balanced including hydrometeorological data and soils surveys from diverse agencies but also streamflow data collected by the research team. I have no objections concerning the methodological approach and I have also found valuable the connections beyond the watershed scale concerning water circulation through faulted geological basements.
I think the paper would have benefit from an English proofreading before submission as I had to suggest a good amount of corrections throughout the manuscript. There are some clarifications requested or suggested too which have to do with results presentation. Although the English style and results suggestions are numerous, I don't think they amount to a major revision of this reseach and they can be easily addressed. I would add a final English proofreading review to make sure everything flows correctly too. I attach the pdf files for the manuscript and supplementary materials with my comments. I inclined to think the paper can be published after a minor review addressed to present some of the results in a clearer way and correct some English writing issues are addressed.
Best regards,
Daniel Blanco-Ward
PD. As I can only attache the pdf for the manuscript, please make sure supplementary is correctly written in the supplementary materials heading and that the the article is correctly written before any of the soil inventories. I think they should add the words soil inventories after those abbreviations too.

Author Response
Dear Reviewer
Thank you for comments. Our response and marked up manuscript are attached
Anwar and Tammo

Reviewer 2 Report
Comments and Suggestions for Authors
The study presents an Assessing digital soil inventories for predicting stream flow in the headwaters of the Blue. Such analysis was performed by SWAT model with different type soil datasets. I consider the main topic of the manuscript is interesting and it is under the scope of the journal. However, some improvements are needed to be performed before considering it for publication. Although results in the manuscript were stated clear, more discussion is needed regarding the available soil data and in context of studies. Moreover, some detailed information is missing about the methodology used. Additional comments as below.
General Comments
What is the main contribution of this study as compared to similar previous studies? Authors could improve the quality of manuscript for reader interest by highlighting what is main problem and knowledge gap they are filling with this paper. For international readers, explain digital soil inventories contribution in streamflow and, what can be learned from this study? The manuscript could be improved by including a figure summarizing the methodology employed in the manuscript. Authors could also emphasize particular strengths and limitations of the study for potential applications of their method in other regions, contexts, and scales.
Specific Comments
Line 62: in in?
Lines 142-143: reference error.
Line 144: reference error.
Line 233. reference error.
Line 161. For international readers, is this public information? Where can it be consulted?
Line 165: Please provide with literature to support the validation of a digital classification of an image, by using images from other sensors.
Line 107. Please provide more detail on how the soil data were processed.
Line 152: Why SWAT model was used? Why not using another tool? It is very important to justify this in the manuscript.
Line 29. Have other methodologies been used to predicting streamflow? What are their advantages and disadvantages? Please include this type of information in the Introduction Section.
Please, highlight the main (innovative) achievements of the research.

Author Response
Dear Reviewer
Thank you for all your comments. They were very helpful. The response and marked up manuscript are attached
Anwar and Tammo

Reviewer 3 Report
My opinion is that the manuscript is interesting and of interest to a broad Hydrology audience. A generally well-written paper with good research topic using available spatial data and the SWAT model to predict streamflow. However, before publication, the manuscript needs below major and minor revisions. My review consists of a couple of issues, which I consider mandatory to be properly addressed in the revised version.
Major Comments:
1. Introduction lines 44-49: too simple citations with lots of references, should provide some details of referenced papers with meaningful results from them.
2. Introduction lines 49-58: provide why authors started this study and which objectives authors pursued with this research, having the clear identifications of this study with previously published works in journals. Maybe lines 53 to 58 can be delete (redundant the following study areas and data).
3. If available, consider inserting another type of models (i.e., hybrid model; a lumped conceptual and distributed feature model) as references with below:
1. DeVantier, A.B.; Feldman, A.D. Review of GIS applications in hydrologic modeling. J. Water Resour. Plan. Manag.
1993, 119, 246-261.
2. Cho, Y.; Engel, B.A.; Merwade, V.M. A spatially distributed Clark's unit hydrograph based hybrid hydrologic model (Distributed-Clark). Hydrolog. Sci. J. 2018, 63, 1519-1539.
4. line 96: where is the Bahir Dar station? Could not see in Figure 1, and why and how this station’s data are used?
5. lines 101 to 106: outlet points for streamflow observations are not clear on the map as well as descriptions.
6. lines 153 to 155: which kind of criteria are used for selecting the threshold of hydrographic segmentations? Why authors used 2,500 and 10 ha for two study watersheds?
7. section 4: some of writings in this section (e.g. lines 361 to 383) should be included into section 3 with the corrected section name as “3. Results and Discussion”. And need to insert a new section of “4. Summary and Conclusions” with writings of lines 350 to 359 (as summary) and lines 384 to 389 (as conclusions). In this case, authors have to provide more constructive discussion, summary, and conclusions for each section, respectively.
8. supplementary materials: provide the firm reasons for selections of “a___, v___, and r___” in model calibration using SWAT-CUP. Again, why authors used these options for each calibration parameters? These also make some uncertainty for model calibration.
Minor Comments:
1. line 73: 1974 and 2612 ïƒ 1,974 and 2,612
2. Need to correct for lots of “Error! Reference source not found” (e.g. lines 143, 144, 223, 239, etc.).
3. Use SI units for “ha”, “a-1”, etc.
4. sub-section 2.2.4: provide required equations for the model performance evaluation.
5. Table 4: why the ranked numbers are over the fourteen in the table even though the used parameters for sensitivity analysis are only fourteen?

Author Response
Dear Reviewer
Thank you for your time and comments. Our response and marked up manuscript are attached
Anwar and Tammo

Round 2
Reviewer 3 Report
Congratulations to authors.
I accept the addressed changes, but your reply about the acceptable type of change for "a__, V__, and r__" is still not clear. If available, provide more clear reason or references for it.